

# miR-605-3p may affect caerulein-induced ductal cell injury and pyroptosis in acute pancreatitis by targeting the DUOX2/NLRP3/NF-κB pathway

Gai Zhang[1], Yuanyuan Zhang[2], Bing Wang[3], Hao Xu[1], Donghui Xie[1] and Zhenli Guo[2]

[1] Department of Emergency Internal Medicine, The First Affiliated Hospital of Wannan Medical College Yijishan Hospital, Wuhu, Anhui, China
[2] Department of Oncology, First Affiliated Hospital, Gannan Medical University, Ganzhou, Jiangxi, China
[3] Department of Emergency Surgery, The First Affiliated Hospital of Wannan Medical College Yijishan Hospital, Wuhu, Anhui, China

Corresponding author
Zhenli Guo, waxpt1984@163.com

## ABSTRACT

Acute pancreatitis (AP) is a sudden-onset disease of the digestive system caused by abnormal activation of pancreatic enzymes. Dual oxidase 2 (DUOX2) has been found to be elevated in the progression of a variety of inflammatory diseases. Therefore, we analyzed the specific roles of DUOX2 in AP development. Blood samples were collected from of AP patients and healthy people, and the caerulein-stimulated human pancreatic duct cells (H6C7) were utilized to establish an AP cell model. Cell growth and apoptosis were measured using an MTT assay and TUNEL staining. Additionally, RT-qPCR and western blot assays were conducted to assess the RNA and protein expressions of the cells. ELISA kits were used to determine TNF-α, IL-6, IL-8, and IL-1β levels. The interaction between DUOX2 and miR-605-3p was predicted using the Targetscan database and confirmed by dual-luciferase report assay. We found that DUOX2 increased while miR-605-3p decreased in the blood of AP patients and caerulein-stimulated H6C7 cells. DUOX2 was targeted by miR-605-3p. Furthermore, DUOX2 knockdown or miR-605-3p overexpression promoted cell viability, decreased the TNF-α, IL-6, IL-8, and IL-1β levels, and inhibited apoptosis rate in caerulein-stimulated H6C7 cells. DUOX2 knockdown or miR-605-3p overexpression also increased the Bcl-2 protein levels and down-regulated Bax, cleaved-caspase-1, NLRP3 and p-p65. Interestingly, DUOX2 overexpression reversed the miR-605-3p mimic function in the caerulein-treated H6C7 cells. In conclusion, our research demonstrated that DUOX2 knockdown relieved the injury and inflammation in caerulein-stimulated H6C7 cells.

## INTRODUCTION

Acute pancreatitis (AP) is a sudden-onset disease of the digestive system caused by the abnormal activation of pancreatic enzymes (*Garg & Singh, 2019*). Mild AP is characterized by nausea, fever, and abdominal pain. When it develops into severe AP, a strong, acute

inflammatory response will prompt the body to release pro-inflammatory factors and increase the secretion of toxic substances in the adenous fluid, thus aggravating the pancreatic tissue (*Banks et al., 2013*; *Oppenlander, Chadwick & Carman, 2022*). AP patients develop high fever, abdominal distension, and shock, which can be life-threatening (*Wilmer, 2004*). This disease presents a significant and unresolved challenge for physicians due to its complex pathology and multifactorial characteristics.

Dual oxidase 2 (DUOX2) is an important member of the NADPH oxidase family; it can resist bacterial invasion and regulate the intestinal flora (*Corcionivoschi et al., 2012*; *Taylor & Tse, 2021*). It has been found to be elevated in the progression of inflammatory diseases such as colitis (*MacFie et al., 2014*) and pneumonia (*Lu, Wu & Yang, 2015*). NADPH oxidase may promote the activation of inflammatory signaling pathways in the pathogenesis of AP (*Shen et al., 2018*; *Wen et al., 2019*), such as the NF-κB signaling pathway (*Razliqi et al., 2023*). However, the role of DUOX2 in AP remains unclear.

MicroRNA (miRNA) belongs to endogenous, functional non-coding RNAs with lengths of 17 to 25 nts (*Mohr & Mott, 2015*). Previous research indicates that miRNAs are closely related to various biological behaviors including metabolism, tumor formation, and inflammatory responses, including apoptosis and cell death (*Jiang et al., 2022*; *Rupaimoole & Slack, 2017*). In recent years, many studies have demonstrated that miRNA is closely related to the AP progression (*Song et al., 2021*; *Xiang et al., 2017*), which has implications for the diagnosis and pathogenesis of AP.

Using the target scan online database, we found that miR-605-3p targeted DUOX2. Therefore, we hypothesized that DUOX2 knockdown may inhibit the inflammation of H6C7 negatively regulating miR-605-3p. In this study, we first collected blood samples from AP patients and healthy people, established AP cell model with human pancreatic duct cells (H6C7) stimulated by caerulein, observed the apoptosis and growth of cells, as well as the expression of RNA and protein, and detected the release level of inflammatory factors and the interaction between DUOX2 and miR-605-3p. We found that DUOX2 targeted miR-605-3p. Then down regulated DUOX2 or overexpressed miR-605-3p. The results showed that DUOX2 knockdown could alleviate the injury and inflammatory response of H6C7 cells stimulated by caerulein, while DUOX2 overexpression reversed the miR-605-3p mimicry function in H6C7 cells treated with caerulein, suggesting that targeting the miR-605-3p/DUOX2 axis may be a potential AP treatment strategy.

## MATERIALS AND METHODS

### Collection of clinical blood samples

Thirty AP patients and 30 healthy persons were recruited for this study. We collected 3 ml of blood from each participant and stored the samples at $-80\,°C$ for subsequent experiments. This study was approved by the ethics committee of the First Affiliated Hospital of Wannan Medical College Yijishan Hospital (No. LLSC-2022-178) and informed consent from all participants.

First, plasma sample preparation was completed within 2 h after blood collection. During blood collection, a disposable sterile syringe will be used to draw blood samples

from the donor and will be stored in containers containing anticoagulants (such as sodium citrate or EDTA) to prevent blood coagulation. Subsequently, the collected blood is put into a centrifuge and spun at a certain speed, so that the red blood cells, white blood cells, platelets and other tangible components sink, while the plasma is located in the upper layer. Next, remove the sediment that may remain in the upper layer. This step can be achieved by sucking a certain amount of plasma and slowly adding an equal volume of normal saline along the container wall. Then, the processed plasma samples are sub packed into sterile tubes or plastic tubes, and the capacity of each tube is generally 1∼2 ml. Finally, the label of each test tube should indicate the source of the sample, the date and time of collection and other information for subsequent tracking and recording.

For long-term storage, the plasma can be put into liquid ammonia, and for short-term storage, the plasma can be put into a refrigerator at −20 °C or −70 °C. For subsequent ELISA and RT qPCR related experiments.

## Cell culture and treatment

The human pancreatic duct cell line (H6C7, ATCC, Manassas, VA, USA) was maintained in DMEM containing 10% FBS for 24 h (37 °C, 5% $CO_2$). According to a previous study (*Ahmad et al., 2020*), the H6C7 cells were stimulated with 100 nM caerulein for 24 h to establish AP cell model.

## Cell transfection

The small interfering RNA DUOX2 (si-DUOX2 #1, si-DUOX2 #2), miR-605-3p mimic, pcDNA3.1-DUOX2, and their negative controls (si-NC, mimic NC, empty pcDNA3.1) were purchased from RiboBio (Guangzhou, China). The transfection of these plasmids was conducted using Lipofectamine™ 3000 (Invitrogen, Carlsbad, CA, USA) for 48 h.

## RNA extraction and RT-qPCR

The RNAs were isolated from clinical samples and cells (including those stably expressing si-DUOX2, pcDNA3.1-DUOX2 or miR-605-3p mimic) using TRIzol reagent (Invitrogen, Carlsbad, CA, USA) according to the manufacturer's instructions, and were reverse-transcribed to cDNA using qPCR RT Master Mix (Takara, Shiga, Japan). The qPCR experiment was carried out using SYBR Green (Takara, Shiga, Japan) on a LightCycler 480 instrument (Roche, Basel, Switzerland). Finally, the relative expression of target genes or miRNA was normalized to GAPDH or U6.

## Enzyme-linked immunosorbent assay

The contents of tumor necrosis factor-α (TNF-α), interleukin-6 (IL-6), IL-8 as well as IL-1β in blood samples and supernatants of H6C7 cells were measured using enzyme-linked immunosorbent assay (ELISA) kits (Beyotime, Shanghai, China) following the manufacturer's instructions.

## Cell viability assay

The MTT cell viability and proliferation assay kit (ScienCell, Carlsbad, CA, USA) was purchased to assess the viability of the chondrocytes. Cells were seeded in 96 wells. The transfected cells were added to 20 µL of MTT solution and incubated for 4 h. Next, 150 µL

of DMSO was used to treat the cells. Finally, cell viability was measured at 490 nm using a MultiskanFC microplate reader (Thermo Fisher Scientific, Waltham, MA, USA).

### Dual luciferase reporter assay

The QuickChange Multiple Site-directed Mutagenesis Kit (Stratagene, Santa Clara, CA, USA) was used with the pGL3 luciferase promoter plasmid (Promega, Madison, Wisconsin, USA) to generate constructs containing wild type (WT) DUOX2 3′-UTR or mutation type (MUT) DUOX2 3′-UTR. Cells were seeded in 96 wells. Afterwards, the designed plasmids were co-transfected into H6C7 cells together with miR-605-3p mimic and mimic-nc. The luciferase activities of the harvested H6C7 were measured after 48 h using the Dual Luciferase Reporter Assay Kit (Promega, Madison, WI, USA).

### TUNEL staining

Cells were seeded in 12 well plates. PBS was used to rinse the cells for TUNEL staining. The cells were then individually fixed and saturated with 4% paraformaldehyde along with 0.1 Triton X-100. Next, the cells were treated with a TUNEL reaction mixture (Roche, Basel, Switzerland) in the dark, followed by DAPI treatment for 10 min. Finally, the TUNEL positive cells were tracked using a fluorescence microscope (Olympus, Tokyo, Japan).

### Western blot

RIPA lysis buffer (Beyotime, Jiangsu, China) was used to separate the total protein from H6C7 cells. The BCA kit (Invitrogen, Waltham, MA, USA) was purchased to determine the protein concentration. SDS-PAGE (Thermo Fisher Scientific, Waltham, MA, USA) was used to isolate proteins. After undergoing constant pressure electrophoresis, the isolated proteins were transferred to the Immobilon-E PVDF membrane (Merck, Darmstadt, Germany). After the membrane was cleaned with PBS solution three times, it was incubated with BSA solution for 2 h at room temperature. The membrane was cultured with the primary antibodies at 4 °C. Then, the membrane was incubated with the second antibody for 2 h at room temperature. After repeated washing with PBST, the blot was visualized using the Enhanced ECL Chemiluminescent Substrate kit (Yeasen Biotechnology, Shanghai, China).

The primary antibodies were purchased from Abcam (Cambridge, UK), including anti-DUOX2 (1:1500), anti-Bax (1:1200), anti-Bcl-2 (1:2000), anti-cleaved-caspase-1 (1:1000), anti-NLRP3 (1:800), anti-p-p65 (1:1500,) and anti-GADPH (1:3000).

### Statistical analysis

Each experiment was repeated three times. All data was expressed as mean $\pm$ SD and were analyzed using SPSS 20.0. The student's *t-test* was used to analyze the comparison between the two groups. A one-way analysis of variance (ANOVA) with Tukey's test was used to analyze among three or more groups. A value of $p < 0.05$ was considered with statistical significance.

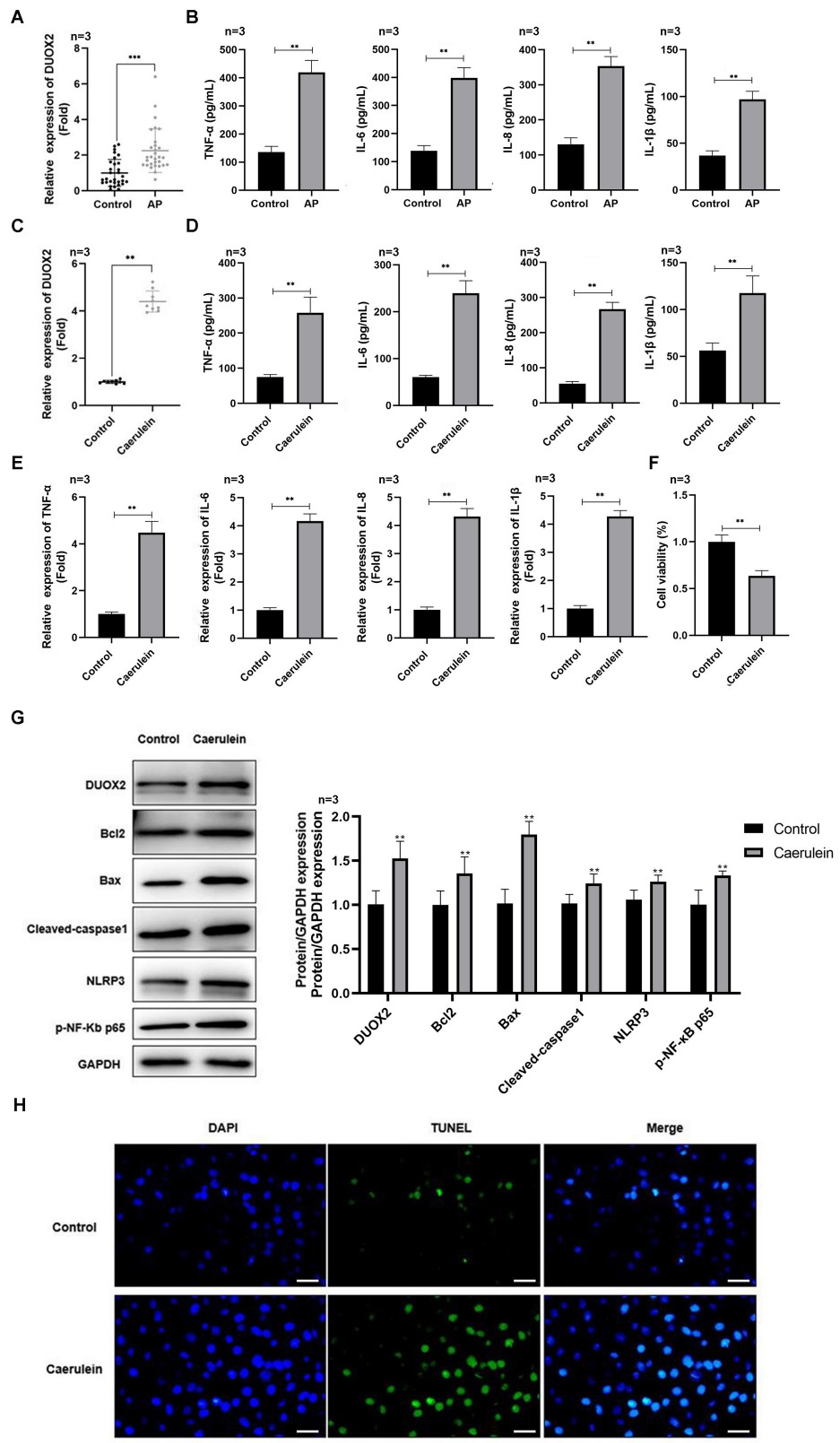

**Figure 1** **DUOX2 was significantly up-regulated in the AP progression.** (A) The DUOX2 levels in the blood samples of AP patients were detected by RT-qPCR assay. (continued on next page…)

**Figure 1 (...continued)**
(B) The contents of TNF-α, IL-6, IL-8, and IL-1β in the blood of AP patients were detected by ELISA kits. (C) The DUOX2 levels in caerulein-treated H6C7 cells were detected by RT-qPCR assay. (D) The contents and (E) mRNA levels of TNF-α, IL-6, IL-8, and IL-1β in the caerulein-treated H6C7 cells were detected by ELISA kits and RT-qPCR assay. (F) The cell viability of caerulein-treated H6C7 cells was tested by MTT assay. (G) The protein levels of DUOX2, Bax, cleaved-caspase-1, NLRP3, p-p65, and Bcl-2 in the caerulein-treated H6C7 cells were determined by western blot. (H) The TUNEL staining was performed to detect the apoptosis of caerulein-treated H6C7 cells (scale bar: 200×). Three biological replicates, $**p < 0.01$, $***p < 0.001$.

# RESULTS

## DUOX2 was significantly up-regulated in AP

DUOX2 was significantly up-regulated in the peripheral blood samples of AP patients *versus* healthy persons (Fig. 1A). Additionally, IL-6, TNF-α, IL-8, and IL-1β was significantly increased in AP patients (Fig. 1B). We treated the human pancreatic duct cell line (H6C7) with caerulein to establish an AP model *in vitro*. We found that DUOX2 was significantly up-regulated in caerulein-treated H6C7 cells (Fig. 1C). The content (Fig. 1D) and mRNA levels (Fig. 1E) of IL-6, TNF-α, IL-8, and IL-1β was also dramatically enhanced in caerulein-treated H6C7 cells, however, cell viability decreased (Fig. 1F). The protein levels of DUOX2, Bax, cleaved-caspase-1, NLRP3, and p-p65 was significantly enhanced and Bcl-2 was down-regulated in caerulein-treated H6C7 cells (Fig. 1G). TUNEL staining results revealed that caerulein treatment significantly increased the apoptosis rate of H6C7 cells (Fig. 1H). The above data showed that DUOX2 and inflammatory expression was up-regulated in the H6C7 cell line model.

## DUOX2 silencing suppressed the inflammation of caerulein-treated H6C7 cells

After si-DUOX2 transfection, the mRNA (Fig. 2A) and protein (Fig. 2B) levels of DUOX2 significantly decreased. Si-DUOX2 #2 was selected for subsequent experiments due to its better knockout efficiency. We found that the contents (Fig. 2C) and mRNA levels (Fig. 2D) of IL-6, TNF-α, IL-8, and IL-1β decreased in the caerulein-treated H6C7 cells following DUOX2 knockdown. In addition, DUOX2 knockdown significantly increased the cell viability of caerulein-treated H6C7 cells (Fig. 2E). The protein levels of Bax, cleaved-caspase-1, NLRP3, and p-p65 decreased and Bcl-2 increased in the caerulein-treated H6C7 cells after DUOX2 knockdown (Fig. 2F). Furthermore, DUOX2 knockdown significantly decreased the apoptosis rate of caerulein-treated H6C7 cells (Fig. 2G). The above data indicate that the expression of DUOX2 promote the development of inflammation and cell death in the H6C7 cell line model.

## MiR-605-3p targeted to DUOX2

Using the Targetscan online database, we determined that miR-605-3p targeted DUOX2 and the 3′UTR of DUOX2 may combine with miR-605-3p (Fig. 3A). miR-605-3p levels significantly decreased in the blood of AP patients (Fig. 3B) and caerulein-treated H6C7 cells (Fig. 3C). The double luciferase report showed that the miR-605-3p mimic significantly

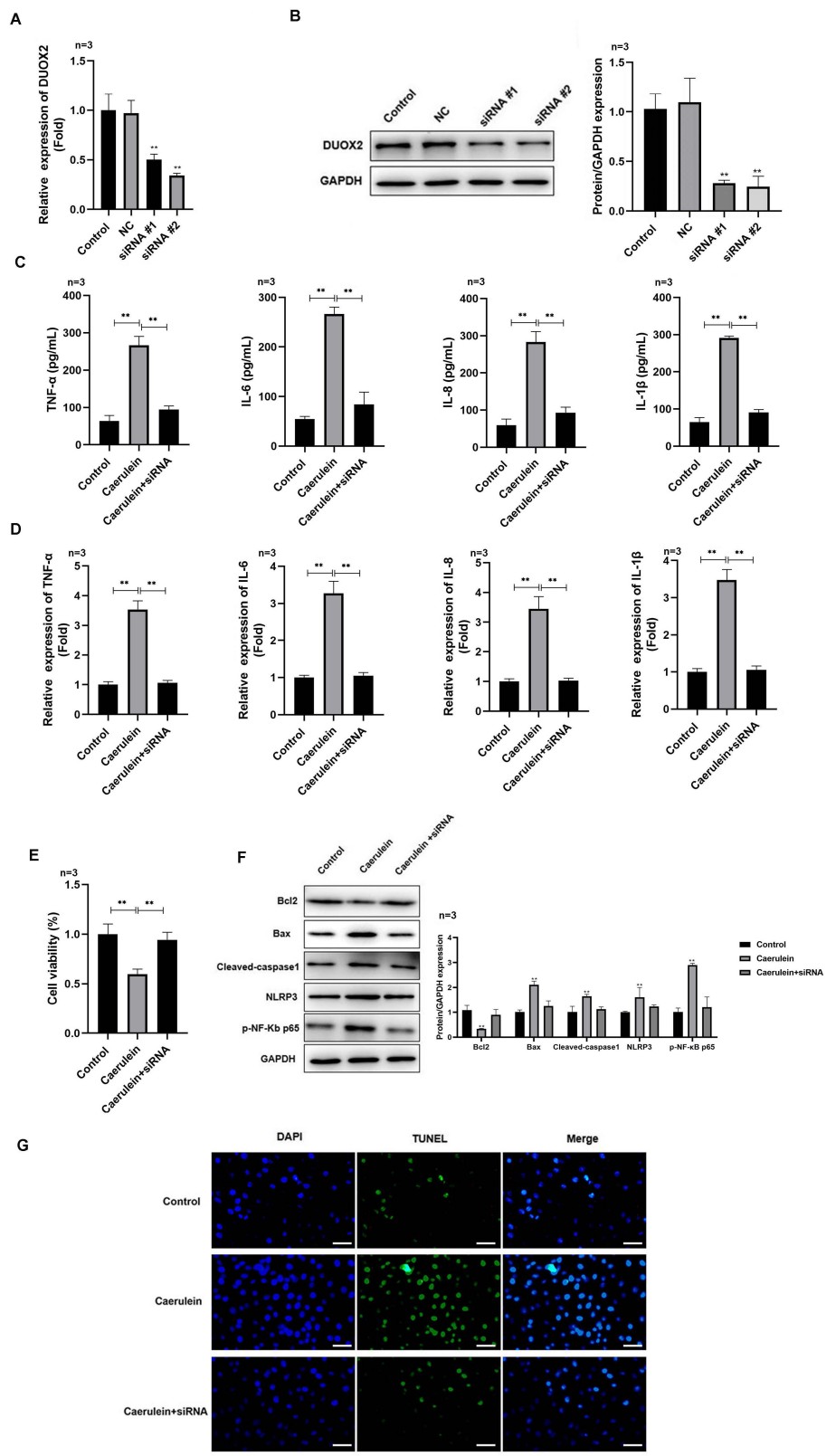

**Figure 2  DUOX2 knockdown inhibited the inflammation of caerulein-treated H6C7 cells.** The knockout efficiency of si-DUOX2 was tested using (A) RT-qPCR and (B) western blot. (continued on next page...)

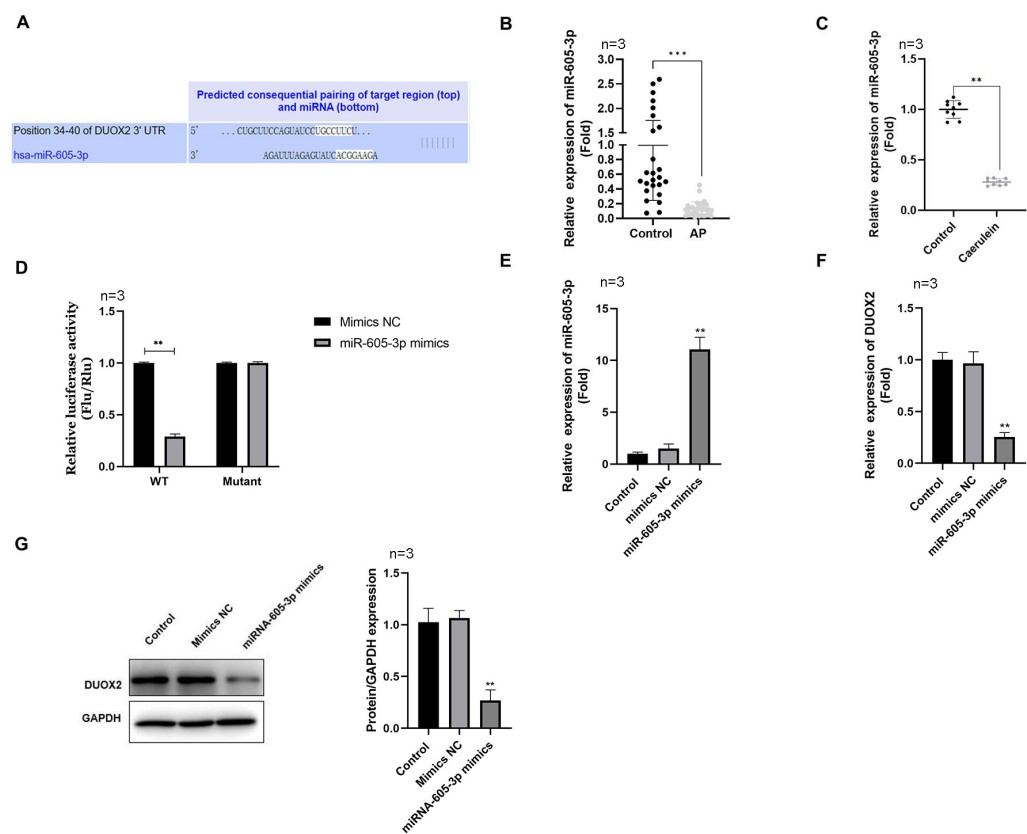

**Figure 3 DUOX2 was a target gene of miR-605-3p.** (A) The binding site between miR-605-3p and DUOX2. (B) The miR-605-3p levels in the blood of AP patients. (C) Caerulein-treated H6C7 cells were detected with RT-qPCR. (D) A double luciferase report was conducted to demonstrate the relationship between miR-605-3p and DUOX2. After miR-605-3p mimic transfection, (E) the miR-605-3p levels were detected by RT-qPCR, the (F) mRNA and (G) protein levels of DUOX2 were measured by western blot. Three biological replicates, **$p < 0.01$, ***$p < 0.001$.

decreased the luciferase activity of WT-DUOX2, and show no effects on that of MUT-DUOX2 (Fig. 3D). Additionally, after miR-605-3p mimic transfection, the miR-605-3p levels significantly increased (Fig. 3E), while the mRNA (Fig. 3F) and protein (Fig. 3G) levels of DUOX2 were significantly down-regulated. The above data showed that the targeted regulation of DUOX2 by miR-605-3p in the H6C7 cell line model.

## DUOX2 overexpression reversed the influence of miR-605-3p mimic in caerulein-treated H6C7 cells

Following pcDNA3.1-DUOX2 transfection, the mRNA (Fig. 4A) and protein (Fig. 4B) levels of DUOX2 were significantly increased. We confirmed that miR-605-3p mimic transfection significantly decreased the content (Fig. 4C) and mRNA levels (Fig. 4D) of IL-6, TNF-α, IL-8, and IL-1β in the caerulein-treated H6C7 cells; conversely, pcDNA3.1-DUOX2 transfection significantly increased them in the caerulein-treated and miR-605-3p mimic transfected H6C7 cells. miR-605-3p mimic transfection increased cell viability (Fig. 5A) and decreased the apoptosis rate (Fig. 5B) of caerulein-treated H6C7 cells, while pcDNA3.1-DUOX2 transfection significantly decreased the cell viability and increased the apoptosis rate of caerulein-treated and miR-605-3p mimic transfected H6C7 cells. Furthermore, Bcl2 protein levels were significantly increased and Bax, cleaved-caspase-1, NLRP3 and p-p65 were significantly decreased in the caerulein-treated H6C7 cells after miR-605-3p mimic transfection. pcDNA3.1-DUOX2 transfection significantly decreased the protein levels of Bcl2 and enhanced the Bax, cleaved-caspase-1, NLRP3, and p-p65 protein levels in caerulein-treated and miR-605-3p mimic transfected H6C7 cells (Fig. 5C). The above data showed that DUOX2 overexpression reversed the influence of miR-605-3p mimic in caerulein- treated H6C7 cells.

## DISCUSSION

AP is a common digestive system disease, which is caused by the abnormal activation of pancreatic enzymes in the pancreas, leading to the self-digestion of pancreatic tissue and inflammation. The expression of DUOX2 is elevated in a variety of inflammatory diseases. Therefore, we analyzed the specific role of Duox2 in AP. Our results showed that DUOX2 was up-regulated in the progression of AP. DUOX2 knockdown inhibited the inflammation in caerulein-treated H6C7 cells by regulating the NLRP3/NF-κB pathway. Mechanistically, miR-605-3p targeted DUOX2, and DUOX2 overexpression reversed the role of miR-605-3p mimic in caerulein-treated H6C7 cells.

Studies have determined that DUOX2 is up-regulated in many cancers, leading to the accumulation of H2O2 and DNA damage through NF-κB signaling (*Wu et al., 2019*; *Wu et al., 2016*). Except in cancer, high levels of DUOX2 caused the release and accumulation of reactive oxygen species, which induced the activation of the pro-inflammatory cytokine, thus leading to inflammation in the body. For example, *Chu et al. (2017)* demonstrated that DUOX2 was highly expressed in the intestinal epithelium. DUOX2 promoted the development of ileocolitis by inducing the secretion of inflammatory factors. Targeting DUOX2 might be a promising therapeutic method to treat ileocolitis. *Wang et al. (2021)* found that DUOX2 knockdown enhanced cell viability and inhibited inflammatory activation by blocking HMGB1 release in the progression of dry eye disease, which suggests that high levels of DUOX2-induced ocular surface inflammation in this disease. Recent research reported that DUOX2 may be a potential prognostic marker in pancreatic cancer (*Cao et al., 2021*). High levels of DUOX2 promoted the malignant behavior of pancreatic cancer cells, such as excessive proliferation

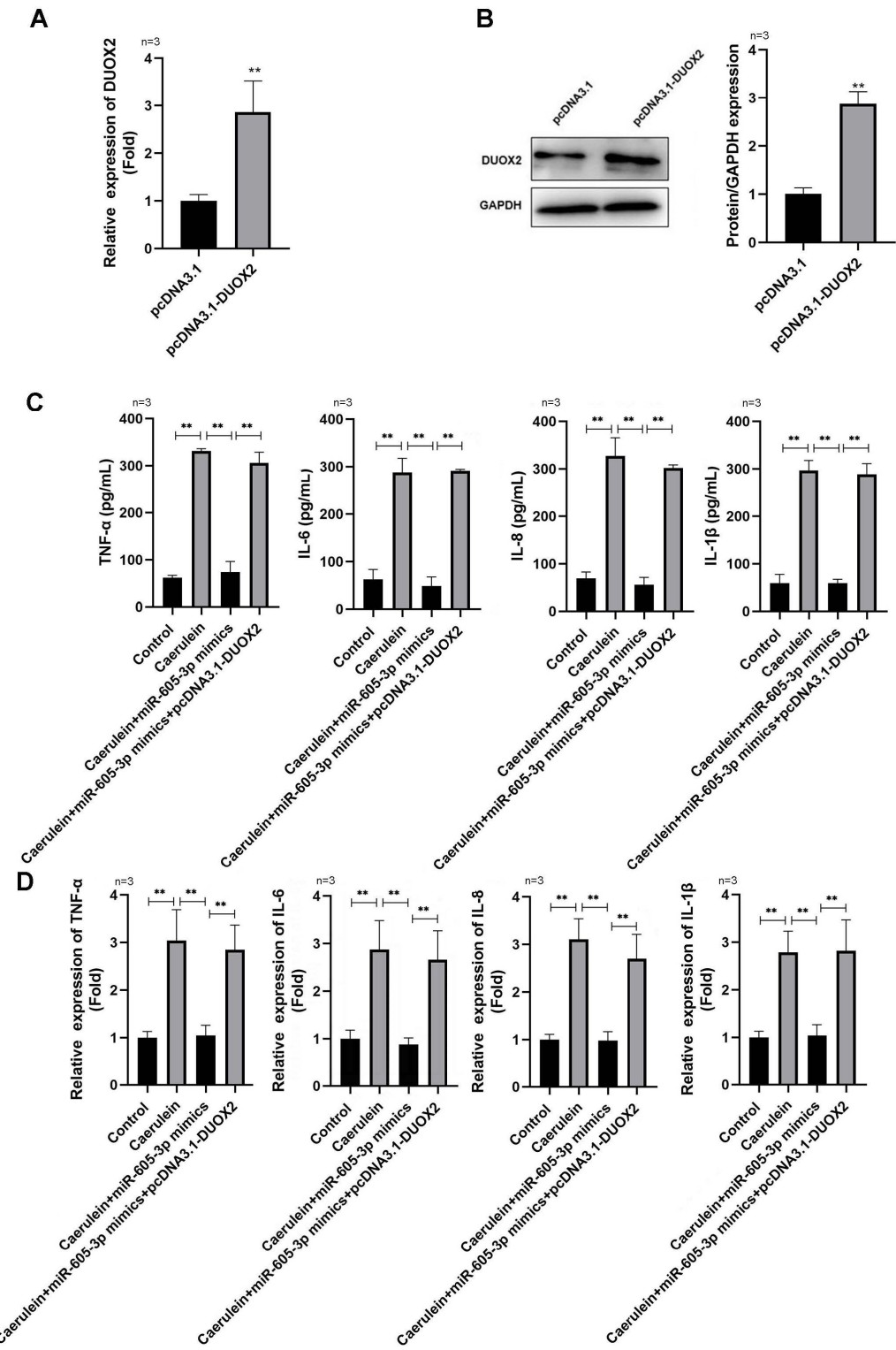

**Figure 4** **DUOX2 overexpression reversed the role of miR-605-3p mimic in caerulein-treated H6C7 cells.** The overexpression efficiency of pcDNA3.1-DUOX2 was tested by (A) RT-qPCR and (B) western blot. (continued on next page...)

**Figure 4 (...continued)**
The caerulein-treated H6C7 cells were transfected with miR-605-3p and pcDNA3.1-DUOX2. Then, (C) the contents and (D) mRNA levels of TNF-α, IL-6, IL-8, and IL-1β were measured by ELISA kits and RT-qPCR assay. Three biological replicates. ** $p < 0.01$.

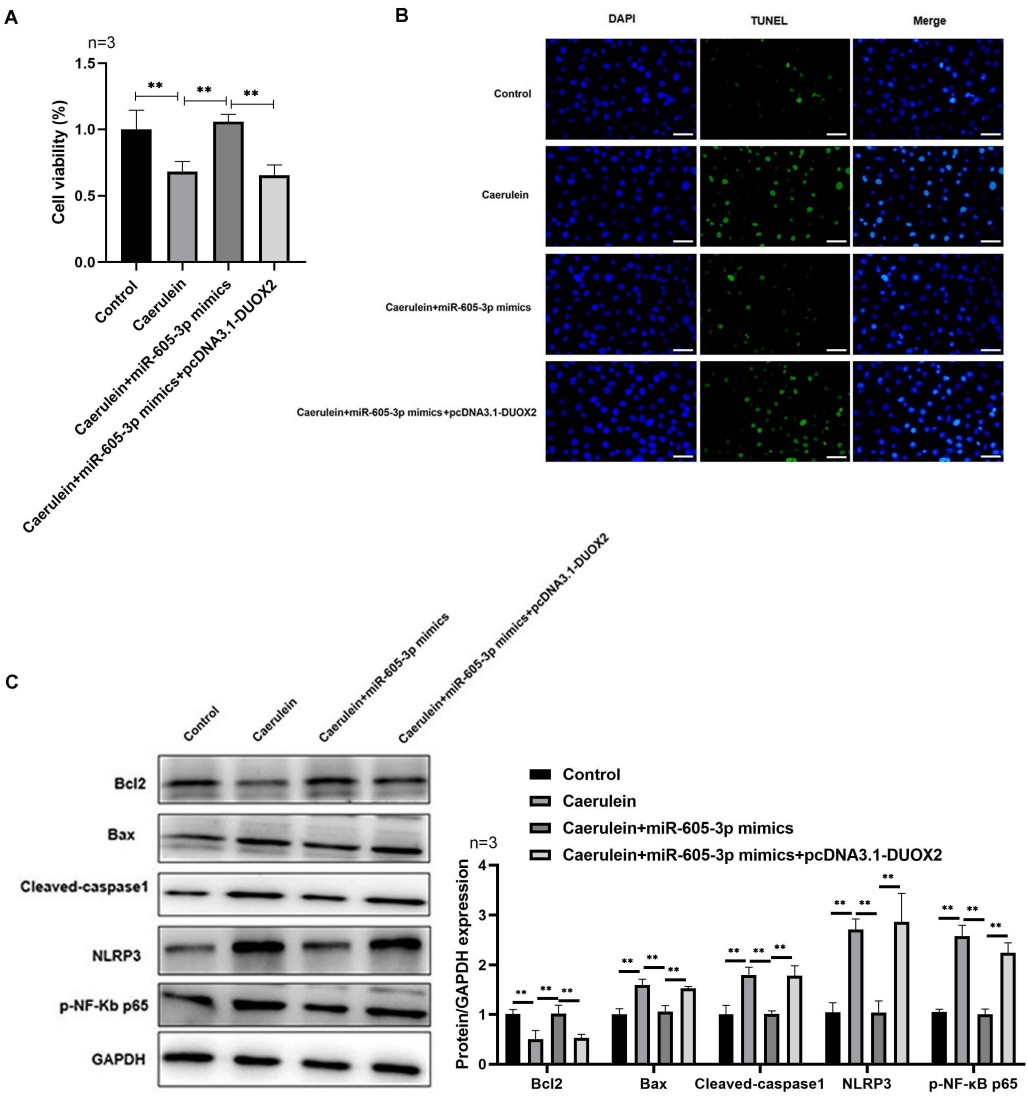

**Figure 5  DUOX2 overexpression reversed the role of miR-605-3p mimic in caerulein-treated H6C7 cells.** (A) The cell viability was tested by MTT assay. (B) The TUNEL staining was performed to detect the apoptosis (scale bar: 200×). (C) The protein levels of Bax, cleaved-caspase-1, NLRP3, p-p65, and Bcl-2 were determined by western blot. ** $p < 0.01$. The above data showed that overexpression of DUOX2 could inhibit the effect of miR-605-3p on mimic in caerulein treated H6C7 cells. Three biological replicates.

(*Wu et al., 2013*), formation of microenvironment promoting angiogenesis (*Wang et al., 2023*) and resistance (*Lyu et al., 2022*). Pancreatitis is closely related to pancreatic cancer (*Vlavcheski et al., 2022*). In clinical practice, about 10% of pancreatic cancer will be misdiagnosed as pancreatitis, and forms of pancreatic cancer are gradually developed from pancreatitis (*Demir, Friess & Ceyhan, 2015*). Therefore, we speculated that the abnormal expression of DUOX2 may be a key factor in the occurrence of AP. However, whether DUOX2 plays an anti-inflammatory role in AP remains unknown. This study found that DUOX2 was up-regulated in the blood sample of AP patients and caerulein-treated H6C7 cells. DUOX2 knockdown inhibited the IL-6, TNF-α, IL-8 as well as IL-1β levels in the caerulein-treated H6C7 cells.

Bax and Bcl2 proteins are apoptosis markers in the process of disease development. Upregulation of Bax and downregulation of Bcl2 have been shown to induce apoptosis (*Edlich, 2018*). Additionally, NLRP3 inflammasome, a member of the protein family of pattern recognition intracellular receptor spot like receptors, is a complex composed of inactive caspase-1 precursor, NLRP3, and ASC (*Kelley et al., 2019*). At rest, NLRP3 expression in cells is extremely low. Once the signal mediating the activation of the NLRP3 inflammasome is activated, the pattern recognition receptor activates NF-κB, leading to the transcription and translation of NLRP3 and pro-IL-1β (*Ding et al., 2016*). In AP progression, damage associated molecular patterns in the pancreas are recognized by the body's immune cells. At the same time, immune cells can also recognize pathogen-associated molecular patterns to induce the activation of NF-κB and inflammasome assembly (*Fawzy et al., 2022*). Activated NF-κB induces the transcription and translation of NLRP3 and pro-IL-1B, completing NLRP3 priming. The function of pro-IL-1B as well as pro-IL-18 further improved under the guidance of activated caspase-1, inducing inflammation in AP (*Wang et al., 2022*). Here, we also found that the protein levels of Bax, cleaved-caspase-1, NLRP3, and p-p65 dramatically declined, while Bcl-2 was enhanced in the caerulein-treated H6C7 cells after DUOX2 knockdown. These results indicated that DUOX2 might participated in AP progression through regulating NLRP3/NF-κB signaling pathway.

Using Targetscan online database, we found that miR-605-3p targeted DUOX2. MiR-605-3p has been shown to influence the development of various cancers by regulating the target gene expression. For instance, *Liu et al. (2019)* determined that miR-605-3p decreased in glioma and suppressed the malignant behaviors of glioma cells *via* negative regulation VASP levels. *Zeng et al. (2019)* found that miR-605-3p levels were lower in bladder cancer cells and miR-605-3p overexpression inhibited the growth and metastasis of bladder cancer cell *via* negative regulation VANGL1 levels. However, the influence of miR-605-3p in AP remains unclear. Here, a double luciferase report was used to confirm the relationship between miR-605-3p and DUOX2. MiR-605-3p overexpression decreased the DUOX2 levels in the caerulein-treated H6C7 cells. DUOX2 overexpression was shown to reverse the effects of miR-605-3p on the inflammatory factors and NLRP3/NF-κB signaling pathway.

In conclusion, this study demonstrated that the miR-605-3p/DUOX2 axis participated in the progression of AP by regulating the NLRP3/NF-κB signaling pathway.

In order to ensure the reliability and accuracy of the results, we will carry out *in vivo* experiments to build and verify the mouse model of acute pancreatitis, and then use gene editing technology to build miR-605-3p or DUOX2 knockout or knock in mouse models, and induce AP on these models, observe the pathogenesis, inflammation and tissue damage of AP, verify that miR-605-3p/DUOX2 axis is involved in the progression of AP, and our results may provide a promising therapeutic strategy for AP in the future.

### Funding

This study was supported by Wuhu Science and Technology Project (No. 2021yf67). The funders had no role in study design, data collection and analysis, decision to publish, or preparation of the manuscript.

### Grant Disclosures

The following grant information was disclosed by the authors:
Wuhu Science and Technology Project: No. 2021yf67.

### Competing Interests

The authors declare there are no competing interests.

### Author Contributions

- Gai Zhang conceived and designed the experiments, prepared figures and/or tables, and approved the final draft.
- Yuanyuan Zhang performed the experiments, prepared figures and/or tables, and approved the final draft.
- Bing Wang performed the experiments, prepared figures and/or tables, and approved the final draft.
- Hao Xu analyzed the data, authored or reviewed drafts of the article, and approved the final draft.
- Donghui Xie analyzed the data, authored or reviewed drafts of the article, and approved the final draft.
- Zhenli Guo analyzed the data, authored or reviewed drafts of the article, and approved the final draft.

### Human Ethics

The following information was supplied relating to ethical approvals (i.e., approving body and any reference numbers):

The First Affiliated Hospital of Wannan Medical College Yijishan Hospital (No. LLSC-2022-178).

### Data Availability

The raw data is available in the Supplemental Files.

## Supplemental Information

Supplemental information for this article can be found online at http://dx.doi.org/10.7717/peerj.17874#supplemental-information.

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
