# Peer review of "miR-605-3p may affect caerulein-induced ductal cell injury and pyroptosis in acute pancreatitis by targeting the DUOX2/NLRP3/NF-κB pathway"

_PeerJ, doi:10.7717/peerj.17874_

## Round 0.1 · original submission · Major Revisions

Based on the comments of two reviewers, I would like you to revise the manuscript, address the scientific and linguistic issues.

**Language Note:** The Academic Editor has identified that the English language must be improved. PeerJ can provide language editing services - please contact us at [email protected] for pricing (be sure to provide your manuscript number and title). Alternatively, you should make your own arrangements to improve the language quality and provide details in your response letter. – PeerJ Staff

Reviewer 1 ·

Basic reporting

The authors set out to study the function of DUOX2 and miR-605-3p in AP development. They tested a variety of cytokines and cell death marker in patient blood and a cell line model at both transcription and expression levels. The research and the writing are straightforward and easy to understand, but there are certain improvements can be made.
1. Line 52 and line 207 both referenced the human thyroid disease. I am confused as to the relationship between the thyroid disease and AP. Why did you specifically mention it. If this piece of information is important to this study, please explain briefly in the corresponding places. Otherwise, I would recommend that you take it out.
2. The last paragraph of the Introduction should be separated into two paragraphs, in my opinion. Because miRNA is an important part in your research, I think the background information on miRNA should have an independent paragraph.
3. The last paragraph of the Introduction should be a brief summary of your research. To be honest, I think you put in too many details in the Abstract that should be here. And a hypothesis is what you propose at the beginning of your research instead of end with a hypothesis. You propose a hypothesis, design experiments to prove or disprove it, reach a conclusion, and discuss future direction.
4. Please replace the word “prominent” with “significant” when your data and statistics show significant differences.
5. In the Discussion, please discuss your future direction based on this research. For example, what’s the next step? is there a mouse model you can use to further test your results in the cell line? how to target the miR-605-3p/DUOX2 axis for therapies?

Experimental design

The experimental design was logical and on point. There are only few minor issues.
1. Please provide more experimental details in the Materials and methods. How were blood samples stored and prepared? Did you use serum or plasma for your ELISA and whole blood for RT-qPCR? As for the cell line experiments, what format was used, 6-well plate, 24-well plates, etc.? Was whole cell lysates or culture supernatant was used in ELISA and RT-qPCR? How many replicates were included in one experiment. How many times did you conduct the experiments to have the final graph and statistics?
2. The western blot images in Figure 1. G, Figure 2. F, Figure 4. F. do not show obvious changes of some of the targets as stated in the Results. Please provide quantified data of the intensity of the bands.
3. Please add the scale bar in the fluorescent images in Figure 1. H, Figure 2. G, and Figure 4. E. In Figure 2. H and Figure 4. E, the IF images are too small to show anything. Please enlarge them to the same size as in Figure 1. H.
4. There is too much blank space in the figures and the labels are too small to read. Please enlarge individual panels and labels accordingly. The labeling in figure 4 is particularly bad. If the labels are too long, please put the individual label on the side and use “+” or “-” directly underneath each group.
5. Please add one or two sentences at the end of each Results section to describe the indication of the data. For example, at the end of the second Results section, you can say something like, the above data indicate that the expression of DUOX2 promote the development of inflammation and cell death in the H6C7 cell line model.

Validity of the findings

No issue here.

Reviewer 2 ·

Basic reporting

General issues:
1. The paper needs a grammar check.
2. There are general problems in placing or removing the (comma) in the sentence.
3. Replace What's more with Furthermore, caerulin treated with caerulin-treated, and caerulin stimulated with caerulin-stimulated.
4. Uniform the word proinflammatory to be pro-inflammatory, both ways are acceptable, but it's best to be consistent.
5. Correct the spaces in the affiliation to be the same as the cover letter as the space before Wuhu in lines 6 and 10.
6. Add (the) before the word Targetscan database, and several words need that correction.
7. Please add the western blot analysis and quantification of the protein levels of all parameters in one figure to clarify and illustrate the figure. Also, the tunnel patterns need detailed descriptions for every group.
Abstract:
1. In line 22, replace was with were to be (were utilized and were measured).
2. In line 27, Remove the word (was) to be miR-605-3p declined in the blood.
Introduction:
In line 41, replace belongs to with is an acute digestive system.
In line 44, add (a) before strong.
In line 55, replace (is) with (has been).
In line 60, replace (researches) with (studies).
In line 62, replace the sentence with (In recent years, many studies have demonstrated that miRNA is closely related to the AP progression).
In line 63, replace (for) with (on).
In line 65, remove to.
In line 68, replace negatively regulating with negative regulation.
Materials and Methods:
1. In line 120, replace (permeate) with (permeated).
2. In line 137, remove the comma after the number 1200 between brackets.
Results:
1. In lines 149 and 152, replace were with was.
2. In line 187, add space between (Figure4C) to be (Figure 4C).
3. In line 196, there is a duplication in the word (the); remove one of them.
Discussion:
In line 208, replace conformed with confirmed.
In line 231, remove (the) before apoptosis.
In line 252, replace was with were.
References: Overall OK.

Experimental design

Methods described in sufficient detail & information.

Validity of the findings

All ok.

Additional comments

Discussion:
Small info should be given before starting the discussion.

---

## Round 0.2 · Minor Revisions

The authors have addressed all of the reviewers' comments and this manuscript is almost ready for publication.

Please could the authors: (i) explain how many technical and biological replicates were performed for every experiment in every panel of every figure. (ii) upload this data as supplementals. e.g. immunoblot of figure 2 file shows only the one blot presented. They must upload all blots from all biological replicates and also the data file of the quantification. (iii) all figures are very small and difficult to read. Submit larger, clearer ones.

Reviewer 1 ·

Basic reporting

The authors resolved all the previous comments.

Experimental design

The authors resolved all the previous comments.

Validity of the findings

The authors resolved all the previous comments.

---

## Round 0.3 · Major Revisions

I have taken over the role of editor for this submission.

We have been very clear on requirements for this work: you must upload as supplemental files all images for ALL immunoblots for ALL biological replicates. Presently you have not done this; you have only provided the blots shown. This must be addressed.

We asked that you explain how many replicates, biological and technical, were performed for each component of each figure within the figure legend. You have not done this.

Similarly you must upload all data which underpins the bar graphs for all figures. This data set cannot be just the means and errors of the replicates, rather it show all the individual experiments, clearly labelled with clear indication of what each column presents. Data transparency is a central tenant of our journal, and this is clearly stated in instructions to authors.

Until this primary data is uploaded and inspected, the submission cannot be accepted.

---

## Round 0.4 · Minor Revisions

We asked that you explain how many replicates, biological and technical, were performed for each component of each figure within the figure legend.

The number is given (e.g. n=3) but this is should be specified as biological or technical replicates for EACH component of EACH figure OR clearly stated as 'X' biological replicates for every panel (or similar).

---

## Round 0.5 · accepted · Accept

Thanks for clarifying the remaining issue. I'm happy to recommend acceptance now.